# Bielectron vortices in two-dimensional Dirac semimetals

C.A. Downing[1,2] & M.E. Portnoi [2,3]

Searching for new states of matter and unusual quasi-particles in emerging materials and especially low-dimensional systems is one of the major trends in contemporary condensed matter physics. Dirac materials, which host quasi-particles which are described by ultra-relativistic Dirac-like equations, are of a significant current interest from both a fundamental and applied physics perspective. Here we show that a pair of two-dimensional massless Dirac–Weyl fermions can form a bound state independently of the sign of the inter-particle interaction potential, as long as this potential decays at large distances faster than Kepler's inverse distance law. This leads to the emergence of a new type of energetically favorable quasiparticle: bielectron vortices, which are double-charged and reside at zero-energy. Their bosonic nature allows for condensation and may give rise to Majorana physics without invoking a superconductor. These novel quasi-particles arguably explain a range of poorly understood experiments in gated graphene structures at low doping.

[1] Université de Strasbourg, CNRS, Institut de Physique et Chimie des Matériaux de Strasbourg, UMR 7504, Strasbourg F-67000, France. [2] School of Physics, University of Exeter, Stocker Road, Exeter EX4 4QL, UK. [3] International Institute of Physics, Universidade Federal do Rio Grande do Norte, Natal—RN 59078-970, Brazil. C.A. Downing and M.E. Portnoi contributed equally to this work. Correspondence and requests for materials should be addressed to C.A.D. (email: downing@ipcms.unistra.fr) or to M.E.P. (email: m.e.portnoi@exeter.ac.uk)

D irac materials have low-energy fermionic excitations described by a Dirac (or Dirac–Weyl in the case of vanishing mass) Hamiltonian. This intriguing property is found in a variety of condensed matter systems, from graphene to d-wave superconductors to the surface states of topological insulators[1]. It follows that analogs to peculiar phenomena previously studied in high-energy physics have now entered the domain of mesoscopic physics. Typical examples include Klein tunnelling[2], Zitterbewegung[3] and atomic collapse[4, 5]. While the most recent development is a hunt for three-dimensional Weyl fermions[6], arguably two-dimensional (2D) Dirac semimetals[7], studied extensively since the exfoliation of graphene, are even more interesting due to their topological non-trivialities and, in the case of graphene, the possibility of manipulating its properties with the help of electrostatic gates. The current focus on quasiparticles in these systems is on fermionic modes, namely: Dirac, Weyl and Majorana fermions. In this article, we consider another type of quasiparticle: charged bosons formed by the pairing of two Dirac–Weyl fermions at the apex of the Dirac cone. At this point the essential chirality of the 2D Weyl quasi-particles is suppressed and quite remarkably their binding becomes possible.

Electrons in 2D Dirac semimetals can be described by the rather exotic single-particle Hamiltonian $H_1 = v_F \, \boldsymbol{\sigma} \cdot \mathbf{p}$, where $v_F$ is the Fermi velocity and $\boldsymbol{\sigma} = (\sigma_x, \sigma_y)$ are the spin matrices of Pauli. Recently, special attention has been paid to the apexes of the resultant conical dispersion (Dirac points) in the band structures of 2D Dirac–Weyl systems, with studies ranging from defect-induced zero-energy states[8–10], which are speculated to be a source of magnetism in graphene, to Majorana zero modes in topological insulators in proximity to superconductors[11]. In this work, we predict theoretically 2D Dirac semimetals as host materials for another type of hitherto overlooked quasiparticle associated with the Dirac point: stationary (zero center-of-mass motion) bielectron vortices. Intriguingly, these bosonic quasiparticles may be a source of a new type of condensate.

Pair formation in graphene was considered in connection to an excitonic insulator, discussed well before the isolation of graphene[12] and revisited thereafter[13–15]. There has been significant interest in spatially-separated double-layer graphene[16–20], where several groups have studied Bose–Einstein condensation[21–23] and superfluidity[24, 25] in this gapped system. However, no gap has as yet been observed experimentally in monolayer graphene structures[26] and a question remains: can two charge carriers bind together in an ideal 2D Dirac–Weyl system? It has previously been claimed that excitons do not exist in gapless graphene[27], and so considerations of trigonal warping, which effectively introduces an angular-dependent single-particle mass, have been suggested as a route toward pair formation[28–30].

It is a commonly held belief that electrostatic confinement of 2D massless Dirac fermions is impossible as a result of the Klein paradox, where there is a perfect transmission for normally incident particles—this prompted proposals for localization via a variety of other means[31–35]. An argument is usually made that conservation of pseudospin $\boldsymbol{\sigma} \cdot \hat{\mathbf{p}} = \pm 1$ forbids bound states in purely electrostatic problems[2]. However, at the Dirac point pseudospin is not well defined, a fact we exploit in this work. Indeed, unlike the case of finite energy, zero-energy bound states may form at the apex of the Dirac cone[36–38]. Mathematically, this is because at finite energy the effective Schrödinger equation at long-range maps on to the problem of scattering states in a non-relativistic system[39]; whereas at zero-energy solutions exist which decay algebraically, depending on the angular momentum quantum number $m$. When $m$ is nonzero the solutions are fully square-integrable, such that they are rotating ring-like states (vortices) avoiding the Klein tunneling due to their vorticity, which results in a nonzero momentum component along the

potential barrier. It should be emphasized that the existence of these fully confined states does not require introducing any effective mass for the quasi-particles via either imposing sublattice asymmetry, which results in a finite band gap, or considering trigonal warping terms. The only requirement for the existence of zero-energy vortices in a strong enough (beyond a critical strength) radially symmetric potential is that it has a long-distance asymptotic decay faster than Coulombic. In practice, the latter condition always takes place in realistic quasi-2D Dirac semimetals due to either screening or, for the case of graphene, the presence of a metallic gate in close proximity to the 2D electron gas (which is necessary to control the carrier density).

Fully confined zero-energy vortices should be clearly distinguished from the widely discussed 'atomic collapse' peculiarities in the graphene density of states in a supercritical attractive Coulomb potential, since the potential decaying as $1/r$ cannot support square-integrable solutions. Notably, the experimentally observed maximum in graphene's density of states in the presence of supercritical impurities[40], which is attributed to the wavefunction collapse, may be also explained using the zero-energy vortices picture in conjunction with optimal screening. Indeed, the observed peaks in the density of states are too close to the Dirac points, and the spatial extent of the measured induced charge density around the impurities is of the order of tens of the graphene lattice constant, which is easier to explain in terms of the large-size vortices rather than the short-scale wavefunction collapse at the impurity center. Furthermore, there has been a recent glut of experiments on electrostatic confinement in graphene[41–46] which may, due to the long lifetimes found, be fingerprints of zero-energy bound states.

In this article, we generalize the principles behind the aforementioned single-particle picture of confinement to the two-body problem. We show that electrostatic binding of same charge particles into bielectron vortices is both possible and energetically favorable, the effects of which will be apparent in local density of states measurements.

## Results

**Model Hamiltonian.** A consideration of two particles with an interaction potential, in the framework of a four-by-four Dirac–Weyl Hamiltonian, shows that at zero energy the sign of the potential is irrelevant for confinement. This is because the interaction potential only appears as a logarithmic derivative or as squared. Thus, forming bielectron vortices is as much a possibility as binding electrons with holes to construct excitons. The binding of repelling particles is a consequence of the symmetric gapless band structure of graphene, such that the negative kinetic energy can fully compensate electrostatic repulsion. The considered bound pairs have to be static, since two particles may only bind if they have a zero total wavevector $\mathbf{K}$; thus we deal with 'pinned' vortex pairs. This is because for a nonzero $\mathbf{K}$ the angular momentum $m$ is no longer a good quantum number, and necessitates one to seek a solution as a linear combination of relative motion wavefunctions with all possible values of $m$. However, this expansion includes the non-square-integrable component corresponding to $m = 0$ which acts to deconfine the whole quantum state.

It is important to consider either screened systems or gated structures, which modifies the interaction from a purely Coulombic potential[47] for which no square-integrable solutions exist. The presence of metallic gates inevitably leads to image charges resulting in a fast interaction decay[48, 49] at large distances and it is reasonable to introduce a cutoff at short range to avoid a Coulombic singularity. Of course, in this setup the dielectric environment is still of great importance[50], as is the geometry of

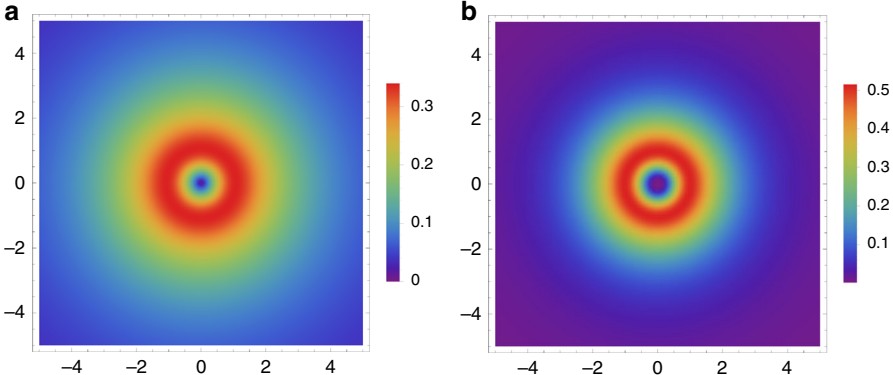

**Fig. 1** Radial probability densities of bielectron vortices A plot of the radial probability densities for the first two bielectron states, with the quantum numbers **a** $(m, n) = (1, 0)$ and **b** $(m, n) = (2, 0)$, as a function of position in two-dimensions. The spatial coordinates are measured in units of the length scale $d$. The color bar measures the dimensionless number associated with the probability density. The interaction strengths are given by Eq. (6)

the device, which both contribute to the effective strength of the interaction. As we demonstrate below, the seemingly rigid conditions on the strength and spatial extent of the inter-particle potential, required to maintain the total energy at zero, are in fact easily satisfied for large-size vortices by linear screening provided by a small number of residual free carriers.

Previous theoretical works on excitonic effects in Dirac materials have approached the problem via either exact diagonalization[51], the Bethe–Salpeter formalism[15, 52] or in the language of a two-body matrix Hamiltonian[53–55], which we will utilize here. The two-body Hamiltonian can be written as the Kronecker sum of the single-particle Hamiltonians $H = H_1 \oplus H_2$, or explicitly (as there are two sublattices and two particles) as the $4 \times 4$ matrix

$$H = v_F \begin{bmatrix} 0 & p_{x_2} - ip_{y_2} & p_{x_1} - ip_{y_1} & 0 \\ p_{x_2} + ip_{y_2} & 0 & 0 & p_{x_1} - ip_{y_1} \\ p_{x_1} + ip_{y_1} & 0 & 0 & p_{x_2} - ip_{y_2} \\ 0 & p_{x_1} + ip_{y_1} & p_{x_2} + ip_{y_2} & 0 \end{bmatrix}, \quad (1)$$

where the subscripts 1 and 2 refer to the two particles. The matrix Hamiltonian given by Eq. (1) is written for two electrons belonging to the same Dirac valley. It can be modified for the particles of different charge (electron and hole) and for two particles belonging to different valleys. Here and in what follows we also neglect spin, which is in principle important as it governs the parity of the relative motion function for the same-valley electrons. However, our immediate aim is to demonstrate the existence of bound states leaving classification of all possible pairs for a future work.

We expect our two-particle continuum theory to be a good approximation to the experimental reality, since in the single-particle picture the theory of zero-energy states[36–38] has successfully predicted confinement effects seen in some recent experiments[41, 44, 46]. In addition, these toy model results from Dirac equations have been shown to be robust to sophisticated numerical experiments on finite sized flakes[56].

**Bielectronic solutions of the model.** The Hamiltonian (1) acts upon a two-particle wavefunction constructed via the Kronecker product $\Psi(\mathbf{r}_1, \mathbf{r}_2) = \psi_i(\mathbf{r}_1) \otimes \psi_j(\mathbf{r}_2)$, where $i, j = (A, B)$. In the absence of an interaction potential $U(\mathbf{r}_1 - \mathbf{r}_2)$, diagonalization of Eq. (1) yields four eigenenergies: $E = \pm v_F \left(p_{x_1}^2 + p_{y_1}^2\right)^{1/2} \pm v_F \left(p_{x_2}^2 + p_{y_2}^2\right)^{1/2}$. As is usual with two-body problems, we utilize the center-of-mass and relative motion coordinates:

$X = (x_1 + x_2)/2$, $Y = (y_1 + y_2)/2$, $x = x_1 - x_2$, $y = y_1 - y_2$. Upon assuming a translationally invariant system, such that the center-of-mass momentum $\hbar\mathbf{K}$ is a constant of motion, one can employ the ansatz $\Psi_i(\mathbf{R}, \mathbf{r}) = \exp(i\mathbf{K} \cdot \mathbf{R})\psi_i(\mathbf{r})$, where the index $i = (1, 2, 3, 4)$ numerates the four components of the wavefunction, which span the two sublattices and two particles. As shown in refs. [53–55] when $\mathbf{K} = 0$ one can rewrite the relative motion Cartesian coordinates $(x, y)$ in polar coordinates $(r, \theta)$, eventually reducing Eq. (1) to a system of three equations only for the transformed radial wavefunction components $\phi_i(r)$,

$$\begin{bmatrix} \frac{U(r)-E}{\hbar v_F} & \partial_r + \frac{m}{r} & 0 \\ 2\left(-\partial_r + \frac{m-1}{r}\right) & \frac{U(r)-E}{\hbar v_F} & 2\left(\partial_r + \frac{m+1}{r}\right) \\ 0 & -\partial_r + \frac{m}{r} & \frac{U(r)-E}{\hbar v_F} \end{bmatrix} \begin{bmatrix} \phi_1(r) \\ \phi_2(r) \\ \phi_3(r) \end{bmatrix} = 0, \quad (2)$$

with $m = 0, \pm 1, \pm 2,...$ and where one can take $\phi_4 = 0$.

Let us now consider a model interaction given by $U(r) = U_0/(1 + (r/d)^2)$, with an on-site energy $U_0$ and the long-range cutoff parameter $d$, which may be related to the separation between the 2D semimetal and the back-gate or to the screening length[48, 49]. This model potential provides a reasonable approximation to the more realistic potential decaying at large distances as $1/r^3$, for details see the Supplemental Information. Notably this functional form is well known in optics as the spatially inhomogeneous Maxwell's fish-eye lens[57], and remarkably is the simplest exactly solvable model, as the square well does not admit a nontrivial solution.

The system of equations (2) can be reduced to a second order differential equation for $\phi_2$ only, which admits an analytical solution for the chosen interaction. This solution is square-integrable only at the Dirac point ($E = 0$). The same is true for any potential decaying faster than the Coulomb potential, so from now on we consider zero-energy states only. Now, when $r \sim 0$, one finds the usual short-range behavior $\phi_2 \sim r^{|m|}$. Meanwhile the asymptotic behavior as $r \to \infty$ is given by the decay $\phi_2 \sim r^{|m|-2\eta_m}$, where $\eta_m = \left(|m| + 1 + \sqrt{m^2 + 1}\right)/2$. Thus, we are motivated to seek a solution of Eq. (2) with the ansatz

$$\phi_2(r) = \frac{(r/d)^{|m|}}{\left(1 + (r/d)^2\right)^{\eta_m}} f(r), \quad (3)$$

where $f(r)$ is a polynomial in $r$ that does not affect the short- and long-range behavior. Upon substituting Eq. (3) into Eq. (2), eliminating $\phi_{1,3}(r)$, and using the new variable $\xi = (r/d)^2$, we

arrive at the following equation for $f(\xi)$:

$$\xi\,(1+\xi)^2 f''(\xi) + (1+\xi)[m+1+(m+2-2\eta_m)\xi]f'(\xi)$$
$$+\left[\left(\frac{1}{4}\frac{U_0 d}{\hbar v_F}\right)^2 - \eta_m^2\right]f(\xi) = 0, \quad (4)$$

which is a form of the Gauss hypergeometric equation[58]. Its solution, regular at $\xi = 0$, is given by

$$f(\xi) = {}_2F_1\left(-n, -n + \frac{1}{2}\frac{|U_0|d}{\hbar v_F}; |m|+1; \frac{\xi}{1+\xi}\right), \quad (5)$$

where we have terminated the power series in the Gauss hypergeometric function ${}_2F_1(a, b; c; x)$ to ensure decaying solutions at infinity. This termination leads to the following quantization condition for the formation of bound bielectron pairs

$$\frac{|U_0|d}{\hbar v_F} = 4(n + \eta_m), \quad n = 0, 1, 2\ldots \quad (6)$$

The other wavefunction components $\phi_{1,3}(r)$ are readily obtainable from Eq. (2), and their long-range behavior $r \to \infty$ tells us that the $m = 0$ state is non-square-integrable, since $(\phi_1, \phi_2, \phi_3) \to r^{-\sqrt{1+|m|^2}}(1, r^{-1}, 1)$. Thus, the pair states are rotating ring-like modes (vortices). Probability density plots are displayed in Fig. 1 for lowest node ($n = 0$) states with $m = 1, 2$. Most noticeable from the figure is the characteristic vortex-like shape of the bielectron states.

Notably, Eq. (6) displays two regimes of interest. In the subcritical case, the threshold value for the first confined state to appear is not met, $\frac{|U_0|d}{\hbar v_F} < \alpha_c$. The critical strength $\alpha_c \simeq 6.83$ is found from Eq. (6) with $n = 0$, $|m| = 1$. However, in the opposing (supercritical) domain $\frac{|U_0|d}{\hbar v_F} \geq \alpha_c$, and pairs may indeed form. Weak screening by a small number of mobile uncoupled carriers allows the system to adjust the inter-particle interaction potential so that it satisfies the strength condition given by Eq. (6) to support bound states, resulting in an energetically favorable drastic reduction in the chemical potential of the many-electron system, accompanied by a narrow spike in the density of states at zero energy. Indeed, pair formation due to doping is a well-known mechanism[59].

**Relation to experiments**. A more accurate treatment of the interaction potential, taking into account both an image charge necessarily present in gated structures and a regularization of the interaction as $r \to 0$, may be tackled numerically (see the Supplemental Information for details). The main difference is that the realistic potential falls at large distances as $1/r^3$, which is faster than the exactly solvable Lorentzian potential. As a result, the critical strength required for binding two electrons depends mostly on the Dirac semimetal fine structure constant (dimensionless interaction strength) $\alpha = e^2/(\kappa \hbar v_F)$ adjusted by a numerical factor of the order of unity, which depends on the ratio of the short-range potential cutoff and its long-range scale. Typical values are $\alpha \simeq 2.19/\kappa, 4.38/\kappa$ for graphene[1] or surface states of three-dimensional topological insulators[60], respectively, where $\kappa$ is the relative permittivity of the material. According to ref.[61], for gapless versions of silicene and germanene $\alpha \simeq 4.06/\kappa, 4.13/\kappa$, respectively. Our numerical estimates (see the Supplementary Information for details) show that the parameter $\alpha_c$ required for forming the first ($|m| = 1$) bielectron vortices is $\alpha_c \simeq 2.5$. This condition is not satisfied for the case of suspended graphene ($\kappa = 1$). However, the discrepancy is not very large and can be compensated by the moderate decrease of the Fermi velocity $v_F$ due to local stretching. Namely, in graphene the local expansion of the honeycomb lattice acts to decrease the

Fermi velocity and thus the effective potential strength may indeed enter the supercritical regime even for this system. In fact, strain-induced corrugations in real graphene samples have been shown to give rise to well-defined regions of electron–hole puddles[62, 63]. Furthermore, the inclusion of static screening[64–66] alone gives the interaction strength for suspended graphene almost sufficient for observing the vortices (see the Supplemental Information for details), and a small additional stretching will help their formation.

Most of the other gapless 2D Dirac systems[1] have Fermi velocities significantly smaller than that of graphene, so the critical strength condition can be easily satisfied. This suggests bielectron vortices should be present at moderate carrier densities in topological insulators, single-valley gapless mercury telluride quantum wells and silicene[67]. Where the Fermi velocity cannot be locally adjusted by stretching, there should be a local pinning of the Fermi level in order to provide optimal screening which maintains the critical interaction strength until the vortices start overlapping (in analogy to a Mott transition). With further carrier density increase, screening effects will lead to the eventual disappearance of vortices when the long-range scale of the potential diminishes. Observing Fermi level pinning with moderate changes of carrier density in low-density 'rigid' Dirac–Weyl systems will be the most unambiguous proof of the existence of bielectron vortices.

One may speculate that fingerprints of bielectron vortices have already been observed in the range of experiments on graphene. Indeed, a reservoir of stationary, zero-energy bielectron vortices may offer a contribution to the experimentally seen Fermi velocity renormalization in gated graphene structures[68–71], which is observed instead of the widely theorized gap. According to this picture, the observed Fermi velocity renormalization could be an artifact of overestimating the number of charge carriers defining the position of the Fermi level; since a large number of them disappear into a many-body ground state of bosonic vortices. The best-known experiment[68] on Fermi velocity renormalization $(v_F \to v_F^*)$ is based on measuring the cyclotron mass, given by $m_c = \hbar(\pi n)^{1/2}/v_F$, where $n$ is the carrier density. However, if a large amount of the carriers condense into a reservoir of zero-energy bosonic vortices, the corrected lower density $n \to n^\star$ of remaining free fermions should be substituted into the cyclotron mass formula, then the smaller observed cyclotron mass may be explained without the need of renormalizing $v_F \to v_F^*$. The same is true for the quantum capacitance measurements[70], since the presence of the charged boson reservoir changes drastically the Fermi energy dependence on the total carrier density from the expected relation, which is used to estimate the renormalized $v_F^*$. Notably, both the original theory of Fermi velocity renormalization in free-standing graphene[72] and its later refinement[73] are based on the long-range behavior of the unscreened Coulomb potential resulting in logarithmically divergent corrections at small $n$. Therefore, we believe that the applicability of these theoretical results should be taken with caution for screened and/or gated structures, in particular graphene on graphite[71].

## Discussion
In conclusion, we have demonstrated that a gapless Dirac–Weyl 2D system with a short-range inter-particle interaction favors the existence of zero-energy charged bound pairs. The associated peak in the local density of states at the Dirac point, which is sensitive to the carrier density, should be taken into account for the interpretation of the scanning tunneling microscopy results. This peak could also serve as a source of carriers with energies corresponding to the strong nonlinear electromagnetic response[74] making low-doped graphene better suited for relevant applications[75].

Arguably, the reservoir of bosonic vortices can play a similar role to that of a superconductor in proximity to a Weyl semimetal by enforcing electron–hole symmetry. Indeed, adding an electron to the considered system is equivalent to adding a hole and another zero-energy vortex which makes this system a promising candidate in the on-going search of Majorana modes in solids. Notably, the particle–hole symmetry provided by the condensate is a necessary, rather than a sufficient, condition for creating Majorana modes. Searching for the most suitable Dirac–Weyl system, which will involve an appropriate Chern number analysis, is one of the avenues for future work.

The observed puddles of charged carriers in graphene in the case of long-range disorder[76] can be treated as many-body mesoscopic domains containing condensates of bosonic bipartite vortices, thus removing the controversy of having carrier puddles despite the absence of single-particle localization in smooth potentials due to the Klein phenomenon. Investigations of this new and unconventional many-body state, with special regard to possible occurrences of quantum critical phase transitions, will form part of a future work. The effect of puddles on the system is controlled by the tiny balance between the electrostatic energy from the positively and negatively charged droplets and the energy of the separating domain walls. The problem is therefore similar to the formation of Landau–Kittel domain structures in ferroelectric materials[77], which was solved recently for domains of an arbitrary shape[78]. The phase diagram of the system with puddle decomposition could even be similar to that discovered in strained dioxyde vanadium, $VO_2$ nanoplatelets with metallic and insulator domain separation, controlled by long-range elastic forces[79].

## Methods
In this theoretical paper, all methods used are fully described in the Results section.

**Data availability**. The authors declare that all of the data supporting the findings of this theoretical study are available within the paper and its supplementary information files.

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

## Acknowledgements

C.A.D. recognizes financial support from the EPSRC DTP (Award reference 1080089). We also acknowledge support from the CNRS, the EU H2020 RISE project CoExAN (Grant No. H2020-644076), EU FP7 ITN NOTEDEV (Grant No. FP7-607521), and the FP7 IRSES projects CANTOR (Grant No. FP7-612285), QOCaN (Grant No. FP7-316432) and InterNoM (Grant No. FP7-612624). We thank E. Mariani, R.J. Nicholas, L.A. Ponomarenko and B.I. Shklovskii for fruitful discussions.

## Additional information

**Competing interests:** The authors declare no competing financial interests.

