## [Peer Review File · Nature Communications]

Reviewers' Comments:

Reviewer #1:

Remarks to the Author:

Authors study the hypothesis of arising of the charged domains (puddles) that can appear in 2D semimetals due to the formation of the Bose statistics electron pairs with their further condensation into Bose condensate. For this, they demonstrate that two electrons can form the bound state, provided that their electrostatic interaction decays more rapidly than Coulomb potential. The subject of research is actual and hot. The hints of such puddles were indeed observed but not satisfactory explained. Hence, the potential impact of the article can be huge, if authors' hypothesis will be proven to be correct.

However, I cannot recommend the publication of this MS in Nature Communication, at least in the present form, by the following reasons.

PRESENTATION.

MS is difficult to read and understand for the broad community, even for physicists, not working in theoretical physics. The text is overloaded by the technical calculations. Its style does not correspond to the recommended format of Nature Communication. The state of art, actuality, and generality of the problem is poorly described. This concerns both the abstract and the introductory part of the text. Importantly, the formulas in many cases, are not preceded by the qualitative explication of the phenomena and the general strategy of calculations. The absence of the graphical presentation makes the reading of the text even more difficult. In general, MS is looking as that, prepared for the more specialized journal.

METHODS

The unclear presentation poses several questions about the validity of the results. Among them are the following:

- Authors claim that the charge bounding doesn't depend on the sign of the interaction. This is not seen from Eq. (6) where the sign of U_0 is important.
- What is the typical size of pairs? Does it smaller than the inter-particle distance? In general, why the two-step procedure of the electron pairing (first) and their condensation (second) is valid, while we deal with the many-body phenomena.
- The bonded pairs will interact electrostatically. Do the 2D Bose condensation is possible?
- The "vortex" terminology for bound pairs with the non-zero momentum, say with $l=1$, can be justified only when the p_x+ip_y state is decoupled from the p_x-ip_y state. Is the degeneracy really removed? What is the reason?

For my opinion, the suitability of the MS for publication in Nature Communication can be possible only after substantial reformatting the article according to the above comments.

Reviewer #2:

Remarks to the Author:

The manuscript by Downing and Portnoi discusses the pair formation of the massless Dirac fermions in two spatial dimensions, which give rise a novel quasi-particle, namely double-charged zero-energy vortices. This work may be relevant, according to the authors, to various two-dimensional Dirac materials.

I found the manuscript interesting, and it may be relevant for the studies of the Dirac materials. However, I think that there are some scientific questions that should be clarified in the manuscript.

First of all, in this paper, α (the coupling constant) is evaluated with the bare values of the parameters. However, there are logarithmic decrease in the coupling constants (as the Coulomb interactions are marginally irrelevant, and the authors consider strictly irrelevant interactions [which decays faster than the Coulomb!]), and so I am not sure if the coupling constant is strong enough at the length scale of the size of the pair. Would this renormalization group picture change

some result in the paper?

Secondly, one of the main claim in the abstract, the realization of the Majorana fermion is not demonstrated/explained much in the manuscript. In particular, the particle-hole symmetry (adding an electron = adding a hole) is not the sufficient condition for Majorana zero mode at the vortex. It requires to clarify, symmetry, index of the BdG Hamiltonian, etc. If the authors want to claim the Majorana fermion to appear at the vortex in these Dirac systems, the authors should show that the system has non-trivial index.

Other than these scientific questions, I do not have any objection in accepting the paper to the journal.

Reviewer #3:

Remarks to the Author:

In this paper, the authors discuss a novel type of bosonic quasiparticle which can result from any type of interaction (interacting and repulsive) as long as the interaction decays fast enough with distance. This leads to the formation of zero energy states in gapless Dirac metals. This is an interesting result and the authors claim that it can't explain the experimental observation of the zero-bias peak in graphene. To the best of my knowledge, it is a new result and has not been studied before. Particularly given the large number system which can realize Dirac fermions, this result will be of great interest to the community. The paper is well written and I believe the results are correct. My only comment is regarding the condensation of the bosonic zero energy modes. Is this zero temperature phenomena? Is the condensate preserved against the thermal fluctuations?

In conclusion, I believe this paper can be considered for publication in Nature Communications.

P. Ghaemi

Report of Referee 1

Authors study the hypothesis of arising of the charged domains (puddles) that can appear in 2D semimetals due to the formation of the Bose statistics electron pairs with their further condensation into Bose condensate. For this, they demonstrate that two electrons can form the bound state, provided that their electrostatic interaction decays more rapidly than Coulomb potential. The subject of research is actual and hot. The hints of such puddles were indeed observed but not satisfactory explained. Hence, the potential impact of the article can be huge, if authors' hypothesis will be proven to be correct.

However, I cannot recommend the publication of this MS in Nature Communication, at least in the present form, by the following reasons.

PRESENTATION.

MS is difficult to read and understand for the broad community, even for physicists, not working in theoretical physics. The text is overloaded by the technical calculations. Its style does not correspond to the recommended format of Nature Communication. The state of art, actuality, and generality of the problem is poorly described. This concerns both the abstract and the introductory part of the text. Importantly, the formulas in many cases, are not preceded by the qualitative explication of the phenomena and the general strategy of calculations. The absence of the graphical presentation makes the reading of the text even more difficult. In general, MS is looking as that, prepared for the more specialized journal.

METHODS

The unclear presentation poses several questions about the validity of the results. Among them are the following: - Authors claim that the charge bounding doesn't depend on the sign of the interaction. This is not seen from Eq. (6) where the sign of U_0 is important. - What is the typical size of pairs? Does it smaller than the inter-particle distance? In general, why the two-step procedure of the electron pairing (first) and their condensation (second) is valid, while we deal with the many-body phenomena. - The bonded pairs will interact electrostatically. Do the 2D Bose condensation is possible? - The "vortex" terminology for bound pairs with the non-zero momentum, say with $l = 1$, can be justified only when the $p_x + ip_y$ state is decoupled from the $p_x - ip_y$ state. Is the degeneracy really removed? What is the reason? For my opinion, the suitability of the MS for publication in Nature Communication can be possible only after substantial reformatting the article according to the above comments.

Report of Referee 2

The manuscript by Downing and Portnoi discusses the pair formation of the massless Dirac fermions in two spatial dimensions, which give rise a novel quasi-particle, namely double-charged zero-energy vortices. This work may be relevant, according to the authors, to various two-dimensional Dirac materials.

I found the manuscript interesting, and it may be relevant for the studies of the Dirac materials. However, I think that there are some scientific questions that should be clarified in the manuscript.

First of all, in this paper, α (the coupling constant) is evaluated with the bare values of the parameters. However, there are logarithmic decrease in the coupling constants (as the Coulomb interactions are marginally irrelevant, and the authors consider strictly irrelevant interactions [which decays faster than the Coulomb]!), and so I am not sure if the coupling constant is strong enough at the length scale of the size of the pair. Would this renormalization group picture change some result in the paper?

Secondly, one of the main claim in the abstract, the realization of the Majorana fermion is not demonstrated/explained much in the manuscript. In particular, the particle-hole symmetry (adding an electron = adding a hole) is not the sufficient condition for Majorana zero mode at the vortex. It requires to clarify, symmetry, index of the BdG Hamiltonian, etc. If the authors want to claim the Majorana fermion to appear at the vortex in these Dirac systems, the authors should show that the system has non-trivial index.

Other than these scientific questions, I do not have any objection in accepting the paper to the journal.

Report of Referee 3

In this paper, the authors discuss a novel type of bosonic quasiparticle which can result from any type of interaction (interacting and repulsive) as long as the interaction decays fast enough with distance. This leads to the formation of zero energy states in gapless Dirac metals. This is an interesting result and the authors claim that it can't explain the experimental observation of the zero-bias peak in graphene. To the best of my knowledge, it is a new result and has not been studied before. Particularly given the large number system which can realize Dirac fermions, this result will be of great interest to the community. The paper is well written and I believe the results are correct. My only comment is regarding the condensation of the bosonic zero energy modes. Is this zero temperature phenomena? Is the condensate preserved against the thermal fluctuations?

In conclusion, I believe this paper can be considered for publication in Nature Communications.

Our response to the reports

We are grateful to the three Referees for their timely and thoughtful reports on our manuscript. We have made changes to the manuscript in line with the suggestions of the Referees, which we believe has led to a stronger and more readable paper. In what follows, we respond to all of the issues raised in the reports point-by-point.

Referee 1 Comment 1:

“MS is difficult to read and understand for the broad community, even for physicists, not working in theoretical physics. The text is overloaded by the technical calculations. Its style does not correspond to the recommended format of Nature Communication. The state of art, actuality, and generality of the problem is poorly described. This concerns both the abstract and the introductory part of the text. Importantly, the formulas in many cases, are not preceded by the qualitative explication of the phenomena and the general strategy of calculations. The absence of the graphical presentation makes the reading of the text even more difficult. In general, MS is looking as that, prepared for the more specialized journal.”

Our answer:

We have reformatted the manuscript into the style of Nature Communications, and edited the abstract and introduction to better describe the state-of-the-art and problem at hand. We have modified the text surrounding the equations to give qualitative explanations of what physics is contained in the formulae.

Referee 1 Comment 2:

“Authors claim that the charge bounding does not depend on the sign of the interaction. This is not seen from Eq. (6) where the sign of U_0 is important.”

Our answer:

We have added an absolute value sign to Eq. (6) to highlight that the sign is not important. The insensitivity to the sign of U_0 can be readily seen from Eq. (4), where U_0 only appears as squared.

Referee 1 Comment 3:

“What is the typical size of pairs? Does it smaller than the inter-particle distance?”

Our answer:

The typical size of the pair is of the order of d , the length scale of the problem, c.f. Eq. (3), which may be related to the separation between the 2D semimetal and the back-gate or to the screening length. When the separation between the 2D semimetal and the gate is smaller than the inverse square root of the inter-particle distance, the pair size is evidently smaller than the inter-particle separation. The picture becomes more complicated when the pair size is defined by free-carrier screening. In this case, the screening length is proportional to the inverse distance between free carriers participating in screening, whereas the inter-particle distance is roughly an inverse square root of the total density. However, the expression for the screening length (the inverse Thomas-Fermi screening wavenumber, see Supplementary Information) contains a large denominator, which is a product of the dimensionless interaction strength (exceeding 2 in suspended graphene and higher in other Weyl semimetals) and a numerical factor ranging between 3.6 and 7.2 depending on the spin and valley degeneracy in the considered system. This allows the total density to be up to two orders of magnitude higher than the density of free carriers before the pairs start overlapping (Mott transition).

Referee 1 Comment 4:

“In general, why the two-step procedure of the electron pairing (first) and their condensation (second) is valid, while we deal with the many-body phenomena.”

Our answer:

In this work, we do not discuss the dynamics of Bose condensation but consider the equilibrium state. We show that two quasiparticles can form a stable immobile zero-energy vortex for a critical value of inter-particle repulsion which is maintained by free-carrier screening. Bielectron vortices are bosons, which in principle can form a macroscopic condensate, which requires a proper many-body description, this description is beyond the scope of our paper. In this respect, our study is somewhat similar to Cooper’s initial work on pair formation preceding the full BCS theory of superconductivity. It is indeed tempting to claim that non-interacting 2D bosons with linear dispersion can condense or use an existing theory developed for repulsive quasi-2D interlayer excitons in double quantum wells. However, for gapless Dirac particles, the binding condition requires the total momentum of the pair (calculated with respect to the Dirac point) to be zero. This requires developing a new many-body theory of the condensate of immobile vortices, and we hope that our work on two-electron binding will encourage creating such a theory.

Referee 1 Comment 5:

“The bonded pairs will interact electrostatically. Do the 2D Bose condensation is possible?”

Our answer:

The inter-particle (screened) Coulomb repulsion within the considered pair is essential for binding. Indeed, the repulsion energy compensates exactly the negative kinetic energy of the relative motion and allows the two particles to bind which is only possible at zero energy (at the Dirac point). Vortex formation resulting in the density of states peak at the apex of Dirac cone reduces the total energy of the system via both the reduction of the kinetic (Fermi) energy and effective cancellation of electrostatic repulsion within the pairs. In our work we mostly focus on demonstrating the possibility of particle binding and pay less attention to inter-vortex correlations. In general, the inter-boson repulsion should not preclude their condensation. However, here we have a peculiar case of immobile double-charged vortices. For an ideal clean system at low temperature, the ground state minimizing inter-vortex repulsion should be a Wigner crystal. In the presence of disorder and especially for the case of a 2D membrane (graphene), which allows local stretching (bubbles) helping the coupling constant to reach a critical value, the system should break into charged domains minimizing the total electrostatic plus elastic energy. These domains, containing the puddles of condensate, are mentioned in the manuscript.

Referee 1 Comment 6:

“The “vortex” terminology for bound pairs with the non-zero momentum, say with $l = 1$, can be justified only when the $p_x + ip_y$ state is decoupled from the $p_x - ip_y$ state. Is the degeneracy really removed? What is the reason?”

Our answer:

The procedure of assigning a well-defined angular momentum to a single particle described by a 2D Dirac-Weyl Hamiltonian in a smooth (non-valley-mixing) radial potential is well-established. Indeed, the total angular momentum operator $J_z = -i\hbar\partial_\theta + \hbar\sigma_z/2$ commutes with the single-particle Hamiltonian. This procedure is easily generalised to a two-particle system with zero total momentum, if both particles in a pair have well-defined valley indexes (for a multi-valley case). Since the size of a pair (discussed in our response to Comment 1) significantly exceeds the lattice constant, there is no grounds to believe that the interaction should mix different valleys. In fact, our goal was to demonstrate the possibility of binding in the most “difficult” general case of a single Dirac cone, indeed, valley mixing allows backscattering and makes binding easier. The specific classification of states spanning all possible spin and valley numbers in a pair of bound particles for certain Dirac materials is left for a future, less general study.

Referee 2 Comment 1:

“First of all, in this paper, α (the coupling constant) is evaluated with the bare values of the parameters. However, there are logarithmic decrease in the coupling constants (as the Coulomb interactions are marginally irrelevant, and the authors consider strictly irrelevant interactions (which decays faster than the Coulomb!)), and so I am not sure if the coupling constant is strong enough at the length scale of the size of the pair. Would this renormalization group picture change some result in the paper?”

Our answer:

The exact solution presented in the manuscript produces a ring-like state of size d , which requires a fixed value of $V_0d/\hbar v_F$ to create a bound bielectron, c.f. Eq. (6). However, if the potential decays one power of distance faster the dependence on d disappears and the ability to form a pair depends only on the dimensionless coupling constant. Notably, the Thomas-Fermi limit of the static screening overestimates screening, whereas we have shown that for the Lorentzian interaction of the same short-distance limit the binding is very robust. Therefore, for graphene we are very close to the binding condition even with overestimated screening, and can be helped in reaching this condition by local stretching reducing v_F by a few per cent (much more plausible than the bare Coulomb logarithmic velocity renormalisation). In fact, the even the static Coulomb interaction is somewhat underestimated, as we (and most of the authors in graphene literature) neglect its enhancement due to a high dielectric constant within the 2D layer. For details on this enhancement, see for example the well known work: L. V. Keldysh, “Coulomb interaction in thin semiconductor and semimetal films”, JETP Lett. **29**, 658 (1979).

It is unlikely that the RG approach will significantly change the results, and will only reduce the paper readability for general public. However, from the Referee report it is clear that if the paper, which shows the possibility of electron-electron binding, is published, significant theoretical development (including the use of the RG approach) can be envisaged.

Referee 2 Comment 2:

“Secondly, one of the main claim in the abstract, the realization of the Majorana fermion is not demonstrated/explained much in the manuscript. In particular, the particle-hole symmetry (adding an electron = adding a hole) is not the sufficient condition for Majorana zero mode at the vortex. It requires to clarify, symmetry, index of the BdG Hamiltonian, etc. If the authors want to claim the Majorana fermion to appear at the vortex in these Dirac systems, the authors should show that the system has non-trivial index. ”

Our answer:

The main point of our work is to demonstrate same-charge carrier pairing, using the simplest 2D Dirac-Weyl Hamiltonian, which makes binding away from the Dirac point impossible. This consideration covers the systems, for which Majorana fermions in the presence of superconductor have been predicted using the relevant symmetry analysis. Although, the electron-hole symmetry is not a sufficient condition for the Majorana zero mode appearance, it is a necessary condition. The predicted condensate of zero-energy vortices indeed provides this condition, as adding to it an electron is equivalent to adding a zero-energy pair and a hole. To our belief, the bielectron vortex formation is possible for a plethora of 2D Weyl systems and checking which of these systems should sustain Majorana zero modes when a superconductor is substituted by a condensate of zero-energy bielectron vortices should become a subject of

extensive future theoretical and hopefully experimental studies.

Referee 3 Comment 1:

“My only comment is regarding the condensation of the bosonic zero energy modes. Is this zero temperature phenomena? Is the condensate preserved against the thermal fluctuations?”

Our answer:

The influence of temperature can be seen as follows: the bielectron pairs are only able to form above a critical potential strength α_c . This α_c is dependent on screening in the system through for example the Thomas-Fermi screening wavenumber q_{TF} , as discussed in the Supplementary Information Section 2. With increasing particle density, q_{TF} increases and so does α_c . Hence there is a critical particle density above which α_c is too high, and the bielectron pairs disassociate. In the Supplementary Information Section 3, which was written to address this specific query, we consider the effect of finite temperature on q_{TF} , and are able to conclude that our proposed binding phenomena is robust against nonzero temperature.

List of changes to the manuscript

- We have corrected some typos in the manuscript
- We have modified the Reference list to fit within the maximum of 70 allowed references
- We have reformatted the manuscript into the style of Nature Communications
- We have revised our Supplementary Information, in response to comment 1 of Referee 3, including the addition of a new figure
- We have improved the presentation of our manuscript, in response to comment 1 of Referee 1

Reviewers' Comments:

Reviewer #1:

Remarks to the Author:

Authors successfully addressed my scientific concerns and improved the presentation styles. Although some deep questions like e.g. the mechanism of Bose condensation in 2D system with repulsion still require more rigorous treatment, the authors' methodology can be considered as plausible and be challenging for the further research, both theoretical and experimental. Hence, I recommend the publication of the manuscript.

Reviewer #2:

Remarks to the Author:

I believe that the response of the authors to my comments are satisfactory except I would like to suggest the authors to make some changes in wording about Majorana fermions.

I agree that the particle-hole symmetry emergent from the bielectron pairing is a necessary condition but it's not a sufficient condition. However, the wordings in the abstract and the main text seems to be written in the way that, any bi-electron pairing in any Dirac-like system is enough to give rise the Majorana fermions, which is apparently misleading.

I strongly recommend the authors to clarify that the bi-electron pairing alone is not enough, but there may be more conditions to be checked, e.g., indices, bi-electron pairing channels, etc.

Other than this, it is a nice manuscript and I believe it is worth to be published in Nature communications.

Reviewer #3:

Remarks to the Author:

I believe that the authors addressed all the points raised by the referees. I find this result particularly interesting due to the fact that they propose a new mechanism for formation of bound states in gapless system and I believe it can generate many followup works and I recommend its publication in current form.

Response to Referees

Report of Referee 1

Authors successfully addressed my scientific concerns and improved the presentation styles. Although some deep questions like e.g. the mechanism of Bose condensation in 2D system with repulsion still require more rigorous treatment, the authors' methodology can be considered as plausible and be challenging for the further research, both theoretical and experimental. Hence, I recommend the publication of the manuscript.

Report of Referee 2

I believe that the response of the authors to my comments are satisfactory except I would like to suggest the authors to make some changes in wording about Majorana fermions. I agree that the particle-hole symmetry emergent from the bielectron pairing is a necessary condition but it's not a sufficient condition. However, the wordings in the abstract and the main text seems to be written in the way that, any bi-electron pairing in any Dirac-like system is enough to give rise the Majorana fermions, which is apparently misleading. I strongly recommend the authors to clarify that the bi-electron pairing alone is not enough, but there may be more conditions to be checked, e.g., indices, bi-electron pairing channels, etc. Other than this, it is a nice manuscript and I believe it is worth to be published in Nature communications.

Report of Referee 3

I believe that the authors addressed all the points raised by the referees. I find this result particularly interesting due to the fact that they propose a new mechanism for formation of bound states in gapless system and I believe it can generate many followup works and I recommend its publication in current form.

Our response to the reports

We thank all of the referees for their positive assessment of our work.

We agree with Referee 2 that the wording could be improved surrounding our discussion of the Majorana physics. Therefore, we have modified our manuscript accordingly in two places, as recommended by Referee 2.

In the abstract, we now say *“Their bosonic nature allows for condensation and may rise to Majorana physics without invoking a superconductor.”*

In the main text, in the second paragraph in the Discussion section, we now state *“Notably, the particle-hole symmetry provided by the condensate is a necessary, rather than a sufficient, condition for creating Majorana modes. Searching for the most suitable Dirac-Weyl system, which will involve an appropriate Chern number analysis, is one of the avenues for future work.”*

List of changes

- We have reworded our discussion of the Majorana physics, in line with the advise of Referee 2
- We have corrected some typos in the manuscript
- We have added some new references, including the recent experimental work of K.-K. Bai et. al., "Generating nanoscale and atomically-sharp p-n junctions in graphene via monolayer-vacancy-island engineering of Cu surface", preprint at <https://arxiv.org/abs/1705.10952> (2017).
- We have broken into two some long sentences in the main text

Review of the stylistic changes

Abstract

We have reworked the abstract so as to fulfill the style of the journal.

Article templates

We have ensured that we comply with our article templates.

Mathematical terms

We have checked that vectors (like our momentum \mathbf{p}) are typeset in bold without italics and that our subscripts (like our Fermi velocity v_F) are in Roman typeface.

Incorporate Supplementary Information into the main manuscript wherever it might be of benefit to the reader

We have checked that the Supplementary Information only contains details that have been relegated from the main text to avoid unnecessary distractions to the reader.

Figures

We have added titles and revised the captions, including stating that the color bar for probability density is a number without dimension.

References

We have added the final page number to all references.

Supplementary Information

We have appropriately organized the Supplementary Information.